# The challenge of realistic music generation: modelling raw audio at scale

**Sander Dieleman**  **Aäron van den Oord**  **Karen Simonyan**
DeepMind
London, UK
{sedielem,avdnoord,simonyan}@google.com

## Abstract

Realistic music generation is a challenging task. When building generative models of music that are learnt from data, typically high-level representations such as scores or MIDI are used that abstract away the idiosyncrasies of a particular performance. But these nuances are very important for our perception of musicality and realism, so in this work we embark on modelling music in the *raw audio* domain. It has been shown that autoregressive models excel at generating raw audio waveforms of speech, but when applied to music, we find them biased towards capturing local signal structure at the expense of modelling long-range correlations. This is problematic because music exhibits structure at many different timescales. In this work, we explore autoregressive discrete autoencoders (ADAs) as a means to enable autoregressive models to capture long-range correlations in waveforms. We find that they allow us to unconditionally generate piano music directly in the raw audio domain, which shows stylistic consistency across tens of seconds.

## 1 Introduction

Music is a complex, highly structured sequential data modality. When rendered as an audio signal, this structure manifests itself at various timescales, ranging from the periodicity of the waveforms at the scale of milliseconds, all the way to the musical form of a piece of music, which typically spans several minutes. Modelling all of the temporal correlations in the sequence that arise from this structure is challenging, because they span many different orders of magnitude.

There has been significant interest in computational music generation for many decades [11, 20]. More recently, deep learning and modern generative modelling techniques have been applied to this task [5, 7]. Although music can be represented as a waveform, we can represent it more concisely by abstracting away the idiosyncrasies of a particular performance. Almost all of the work in music generation so far has focused on such *symbolic representations*: scores, MIDI[1] sequences and other representations that remove certain aspects of music to varying degrees.

The use of symbolic representations comes with limitations: the nuances abstracted away by such representations are often musically quite important and greatly impact our enjoyment of music. For example, the precise timing, timbre and volume of the notes played by a musician do not correspond exactly to those written in a score. While these variations may be captured symbolically for some instruments (e.g. the piano, where we can record the exact timings and intensities of each key press [41]), this is usually very difficult and impractical for most instruments. Furthermore, symbolic representations are often tailored to particular instruments, which reduces their generality and implies that a lot of work is required to apply existing modelling techniques to new instruments.

## 1.1 Raw audio signals

To overcome these limitations, we can model music in the raw audio domain instead. While digital representations of audio waveforms are still lossy to some extent, they retain all the musically relevant information. Models of audio waveforms are also much more general and can be applied to recordings of any set of instruments, or non-musical audio signals such as speech. That said, **modelling musical audio signals is much more challenging than modelling symbolic representations, and as a result, this domain has received relatively little attention**.

Building generative models of waveforms that capture musical structure at many timescales requires high representational capacity, distributed effectively over the various musically-relevant timescales. Previous work on music modelling in the raw audio domain [10, 13, 31, 43] has shown that capturing local structure (such as timbre) is feasible, but capturing higher-level structure has proven difficult, even for models that should be able to do so in theory because their receptive fields are large enough.

## 1.2 Generative models of raw audio signals

Models that are capable of generating audio waveforms directly (as opposed to some other representation that can be converted into audio afterwards, such as spectrograms or piano rolls) are only recently starting to be explored. This was long thought to be infeasible due to the scale of the problem, as audio signals are often sampled at rates of 16 kHz or higher.

Recent successful attempts rely on autoregressive (AR) models: WaveNet [43], VRNN [10], WaveRNN [23] and SampleRNN [31] predict digital waveforms one timestep at a time. WaveNet is a convolutional neural network (CNN) with dilated convolutions [47], WaveRNN and VRNN are recurrent neural networks (RNNs) and SampleRNN uses a hierarchy of RNNs operating at different timescales. For a sequence $x_t$ with $t = 1, \ldots, T$, they model the distribution as a product of conditionals: $p(x_1, x_2, \ldots, x_T) = p(x_1) \cdot p(x_2|x_1) \cdot p(x_3|x_1, x_2) \cdot \ldots = \prod_t p(x_t|x_{<t})$. AR models can generate realistic speech signals, and despite the potentially high cost of sampling (each timestep is produced sequentially), these models are now used in practice for text-to-speech (TTS) [45]. An alternative approach that is beginning to be explored is to use Generative Adversarial Networks (GANs) [16] to produce audio waveforms [13].

Text-to-speech models [23, 43] use strong conditioning signals to make the generated waveform correspond to a provided textual input, which relieves them from modelling signal structure at timescales of more than a few hundred milliseconds. Such conditioning is not available when generating music, so this task requires models with much higher capacity as a result. SampleRNN and WaveNet have been applied to music generation, but in neither case do the samples exhibit interesting structure at timescales of seconds and beyond. These timescales are crucial for our interpretation and enjoyment of music.

## 1.3 Additional related work

Beyond raw audio generation, several generative models that capture large-scale structure have been proposed for images [2, 3, 12, 27], dialogue [40], 3D shapes [30] and symbolic music [37]. These models tend to use hierarchy as an inductive bias. Other means of enabling neural networks to learn long-range dependencies in data that have been investigated include alternative loss functions [42] and model architectures [1, 8, 14, 19, 28, 29, 32]. Most of these works have focused on recurrent models.

## 1.4 Overview

We investigate how we can model long-range structure in musical audio signals efficiently with autoregressive models. We show that it is possible to model structure across roughly 400,000 timesteps, or about 25 seconds of audio sampled at 16 kHz. This allows us to generate samples of piano music that are stylistically consistent. We achieve this by hierarchically splitting up the learning problem into separate stages, each of which models signal structure at a different scale. The stages are trained separately, mitigating hardware limitations. Our contributions are threefold:

- We address music generation in the raw audio domain, a task which has received little attention in literature so far, and establish it as a useful benchmark to determine the ability of a model to capture long-range structure in data.

- We investigate the capabilities of autoregressive models for this task, and demonstrate a computationally efficient method to enlarge their receptive fields using *autoregressive discrete autoencoders* (ADAs).

- We introduce the *argmax autoencoder* (AMAE) as an alternative to vector quantisation variational autoencoders (VQ-VAE) [46] that converges more reliably when trained on a challenging dataset, and compare both models in this context.

## 2 Scaling up autoregressive models for music

To model long-range structure in musical audio signals, we need to enlarge the receptive fields (RFs) of AR models. We can increase the RF of a WaveNet by adding convolutional layers with increased dilation. For SampleRNN, this requires adding more *tiers*. In both cases, the required model size grows logarithmically with RF length. This seems to imply that scaling up these models to large RFs is relatively easy. However, these models need to be trained on audio excerpts that are at least as long as their RFs, otherwise they cannot capture any structure at this timescale. This means that the memory requirements for training increase linearly rather than logarithmically. Because each second of audio corresponds to many thousands of timesteps, we quickly hit hardware limits.

Furthermore, these models are strongly biased towards modelling local structure. This can more easily be seen in image models, where the RF of e.g. a PixelCNN [44] trained on images of objects can easily contain the entire image many times over, yet it might still fail to model large-scale structure in the data well enough for samples to look like recognisable objects. Because audio signals tend to be dominated by low-frequency components, this is a reasonable bias: to predict a given timestep, the recent past of the signal will be much more informative than what happened longer ago. However, this is only true up to a point, and we will need to redistribute some model capacity to capture long-term correlations across many seconds of audio. As we will see, this trade-off manifests itself as a reduction in signal fidelity, but an improvement in terms of musicality.

### 2.1 Stacking autoregressive models

One way of making AR models produce samples with long-range structure is by providing a rich conditioning signal. This notion forms the basis of our approach: we turn an AR model into an autoencoder by attaching an encoder to *learn* a high-level conditioning signal directly from the data. We can insert temporal downsampling operations into the encoder to make this signal more coarse-grained than the original waveform. The resulting autoencoder then uses its AR decoder to model any local structure that this compressed signal cannot capture. We refer to the ratio between the sample rates of the conditioning signal and the input as the *hop size* ($h$).

Because the conditioning signal is again a sequence, we can model this with an AR model as well. Its sample rate is $h$ times lower, so training a model with an RF of $r$ timesteps on this representation results in a corresponding RF of $h \cdot r$ in the audio domain. This two-step training process allows us to build models with RFs that are $h$ times larger than we could before. Of course, the value of $h$ cannot be chosen arbitrarily: a larger hop size implies greater compression and more loss of information.

As the AR models we use are probabilistic, one could consider fitting the encoder into this framework and interpreting the learnt conditioning sequence as a series of probabilistic latent variables. We can then treat this model as a *variational autoencoder* (VAE) [25, 36]. Unfortunately, VAEs with powerful decoders suffer from *posterior collapse* [6, 9, 18, 46]: they will often neglect to use the latent variables at all, because the regularisation term in the loss function explicitly discourages this. Instead, we choose to remove any penalty terms associated with the latent variables from our models, and make their encoders deterministic in the process. Alternatively, we could limit the RF of the decoders [9, 18], but we find that this results in very poor audio fidelity. Other possibilities include using associative compression networks [17] or the 'free bits' method [26].

## 3 Autoregressive discrete autoencoders

In a typical deterministic autoencoder, the information content of the learnt representation is limited only by the capacity of the encoder and decoder, because the representation is real-valued (if they are very nonlinear, they could compress a lot of information into even a single scalar value) [15]. Instead, we make this representation discrete so that we can control its information content directly.

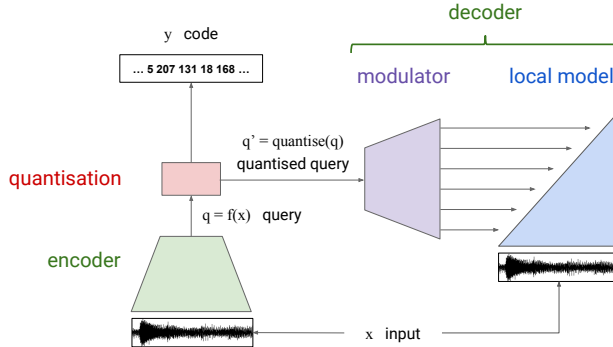

Figure 1: Schematic overview of an autoregressive discrete autoencoder. The encoder, modulator and local model are neural networks.

This has the additional benefit of making the task of training another AR model on top much easier: most succesful AR models of images and audio to date have also treated them as discrete objects [39, 43, 44]. Moreover, we have found in preliminary experiments that deterministic continuous autoencoders with AR decoders may not learn to use the autoregressive connections properly, because it is easier to pass information through the encoder instead (this is the opposite problem of posterior collapse in VAEs).

Figure 1 shows a schematic diagram of an autoregressive discrete autoencoder (ADA) for audio waveforms. It features an *encoder*, which computes a higher-level representation of the input **x**, and a *decoder*, which tries to reconstruct the original waveform given this representation. Additionally, it features a *quantisation* module that takes the continuous encoder output, which we refer to as the *query* vector ($\mathbf{q}$), and quantises it. The decoder receives the quantised query vector ($\mathbf{q}'$) as input, creating a discrete bottleneck. The quantised query can also be represented as a sequence of integers, by using the indices of the quantisation centroids. This is the *code* sequence ($\mathbf{y}$). The decoder consists of an autoregressive *local model* and a *modulator* which uses the quantised query to guide the local model to faithfully reproduce the original waveform. The latter is akin to the conditioning stack in WaveNet for text-to-speech [43]. We will now discuss two instantiations of ADAs.

### 3.1 VQ-VAE

Vector quantisation variational autoencoders [46] use *vector quantisation* (VQ): the queries are vectors in a $d$-dimensional space, and a codebook of $k$ such vectors is learnt on the fly, together with the rest of the model parameters. The loss function takes the following form:

$$\mathcal{L}_{VQ-VAE} = -\log p(\mathbf{x}|\mathbf{q}') + (\mathbf{q}' - [\mathbf{q}])^2 + \beta \cdot ([\mathbf{q}'] - \mathbf{q})^2. \tag{1}$$

Square brackets indicate that the contained expressions are treated as constant w.r.t. differentiation[2]. The three terms are the *negative log likelihood* (NLL) of **x** given the quantised query $\mathbf{q}'$, the *codebook loss* and the *commitment loss* respectively. Instead of minimising the combined loss by gradient descent, the codebook loss can also be minimised using an exponentially smoothed version of the K-means algorithm. This tends to speed up convergence, so we adopt this practice here. We denote the coefficient of the exponential moving average update for the codebook vectors as $\alpha$. Despite its name (which includes 'variational'), no cost is associated with using the encoder pathway in a VQ-VAE.

We have observed that VQ-VAEs trained on challenging (i.e. high-entropy) datasets often suffer from *codebook collapse*: at some point during training, some portion of the codebook may fall out of use and the model will no longer use the full capacity of the discrete bottleneck, leading to worse likelihoods and poor reconstructions. The cause of this phenomenon is unclear, but note that K-means and Gaussian mixture model training algorithms can have similar issues. We find that we can mitigate this to some extent by using *population based training* (PBT) [22] to adapt the hyperparameters $\alpha$ and $\beta$ online during training (see Section 4).

## 3.2 AMAE

Because PBT is computationally expensive, we have also tried to address the codebook collapse issue by introducing an alternative quantisation strategy that does not involve learning a codebook. We name this model the *argmax autoencoder* (AMAE). Its encoder produces $k$-dimensional queries, and features a nonlinearity that ensures all outputs are on the $(k-1)$-simplex. The quantisation operation is then simply an $\mathrm{argmax}$ operation, which is equivalent to taking the nearest $k$-dimensional *one-hot* vector in the Euclidean sense.

The projection onto the simplex limits the maximal quantisation error, which makes the gradients that pass through it (using straight-through estimation [4]) more accurate. To make sure the full capacity is used, we have to add an additional *diversity* loss term that encourages the model to use all outputs in equal measure. This loss can be computed using batch statistics, by averaging all queries $\mathbf{q}$ (before quantisation) across the batch and time axes, and encouraging the resulting vector $\bar{\mathbf{q}}$ to resemble a uniform distribution.

One possible way to restrict the output of a neural network to the simplex is to use a $\mathrm{softmax}$ nonlinearity. This can be paired with a loss that maximises the entropy of the average distribution across each batch: $\mathcal{L}_{diversity} = -H(\bar{\mathbf{q}}) = \sum_i \bar{\mathbf{q}}_i \log \bar{\mathbf{q}}_i$. However, we found that using a ReLU nonlinearity [33] followed by divisive normalisation[3], paired with an $L_2$ diversity loss, $\mathcal{L}_{diversity} = \sum (k \cdot \bar{\mathbf{q}} - 1)^2$, tends to converge more robustly. We believe that this is because it enables the model to output exact zero values, and one-hot vectors are mostly zero. The full AMAE loss is then:

$$\mathcal{L}_{AMAE} = -\log p(\mathbf{x}|\mathbf{q}') + \nu \cdot \mathcal{L}_{diversity}. \tag{2}$$

We also tried adopting the commitment loss term from VQ-VAE to further improve the accuracy of the straight-through estimator, but found that it makes no noticeable difference in practice. As we will show, an AMAE usually slightly underperforms a VQ-VAE with the same architecture, but it converges much more reliably in settings where VQ-VAE suffers from codebook collapse.

## 3.3 Architecture

For the three subnetworks of the ADA (see Figure 1), we adopt the WaveNet [43] architecture, because it allows us to specify their RFs exactly. The encoder has to produce a query sequence at a lower sample rate, so it must incorporate a downsampling operation. The most computationally efficient approach is to reduce this rate in the first few layers of the encoder. However, we found it helpful to perform mean pooling at the output side instead, to encourage the encoder to learn internal representations that are more invariant to time shifts. This makes it harder to encode the local noise present in the input; while an unconditional AR model will simply ignore any noise because it is not predictable, an ADA might try to 'copy' noise by incorporating this information in the code sequence.

## 4 Experiments

To evaluate different architectures, we use a dataset consisting of 413 hours of recorded solo piano music (see appendix B for details). We restrict ourselves to the single-instrument setting because it reduces the variety of timbres in the data. We chose the piano because it is a polyphonic instrument for which many high-quality recordings are available.

Because humans recognise good sounding music intuitively, without having to study it, we can efficiently perform a qualitative evaluation. Unfortunately, quantitative evaluation is more difficult. This is more generally true for generative modelling problems, but metrics have been proposed in some domains, such as the Inception Score [38] and Frechet Inception Distance [21] to measure the realism of generated images, or the BLEU score [34] to evaluate machine translation results. So far, no such metric has been developed for music. We have tried to provide some metrics, but ultimately we have found that listening to samples is essential to meaningfully compare models. We are therefore sharing samples, and we urge the reader to listen and compare for themselves. They can be found at https://bit.ly/2IPXoDu. Most samples are 10 seconds long, but we have also included some minute-long samples for the best unconditional model (#3.6 in Table 3), to showcase

Table 1: Results for ADAs trained on audio waveforms, for different input representations and hop sizes. NLLs are reported in nats per timestep. The NLL of a large unconditional WaveNet model is included for comparison purposes. The asterisks indicate the architectures which we selected for further experiments. † The perplexity for model #1.7 is underestimated because the length of the evaluation sequences was limited to one second.

| # | MODEL | INPUT FORMAT | HOP SIZE | RECONSTRUCTION NLL TRAIN | EVAL | CODEBOOK PERPLEXITY |
|---|---|---|---|---|---|---|
| 1.1 | WaveNet | continuous | N/A | 1.192 | 1.151 | N/A |
| *1.2 | VQ-VAE | continuous | 8 | 0.766 | 0.734 | 227.3 |
| 1.3 | VQ-VAE | one-hot | 8 | 0.711 | 0.682 | 199.6 |
| 1.4 | AMAE | continuous | 8 | 0.786 | 0.759 | 228.9 |
| 1.5 | AMAE | one-hot | 8 | 0.721 | 0.695 | 225.6 |
| 1.6 | AMAE with softmax | one-hot | 8 | 0.833 | 0.806 | 183.2 |
| *1.7 | VQ-VAE | continuous | 64 | 1.203 | 1.172 | 84.04† |

its ability to capture long-range structure. Some excerpts from real recordings, which were used as input to sample reconstructions, are also included.

To represent the audio waveforms as discrete sequences, we adopt the setup of the original WaveNet paper [43]: they are sampled at 16 kHz and quantised to 8 bits (256 levels) using a logarithmic ($\mu$-law) transformation. This introduces some quantisation noise, but we have found that increasing the bit depth of the signal to reduce this noise dramatically exacerbates the bias towards modelling local structure. Because our goal is precisely to capture long-range structure, this is undesirable.

## 4.1 ADA architectures for audio

First, we compare several ADA architectures trained on waveforms. We train several VQ-VAE and AMAE models with different input representations. The models with a *continuous* input representation receive audio input in the form of real-valued scalars between $-1$ and $1$. The models with *one-hot* input take 256-dimensional one-hot vectors instead (both on the encoder and decoder side, although we found that it only makes a difference for the former in practice). Unless otherwise specified, the AMAE models use ReLU followed by divisive normalisation. Details about the model architectures and training can be found in appendix A.

In Table 1, we report the NLLs (in nats per timestep[4]) and codebook perplexities for several models. The perplexity is the exponential of the code entropy and measures how efficiently the codes are used (higher values are better; a value of 256 would mean all codes are used equally often). We report NLLs on training and held-out data to show that the models do not overfit. As a baseline, we also train a powerful unconditional WaveNet with an RF of 384 ms and report the NLL for comparison purposes. Because the ADAs are able to encode a lot of information in the code sequence, we obtain substantially lower NLLs with a hop size of 8 (#1.2 – #1.5) – but these models are conditional, so they cannot be compared fairly to unconditional NLLs (#1.1). Empirically, we also find that models with better NLLs do not necessarily perform better in terms of perceptual reconstruction quality.

**Input representation**   Comparing #1.2 and #1.3, we see that this significantly affects the results. Providing the encoder with one-hot input makes it possible to encode precise signal values more accurately, which is rewarded by the multinomial NLL. Perceptually however, the reconstruction quality of #1.2 is superior. It turns out that #1.3 suffers from partial codebook collapse.

**VQ-VAE vs. AMAE**   AMAE performs slightly worse (compare #1.2 and #1.4, #1.3 and #1.5), but codebook collapse is not an issue, as indicated by the perplexities. The reconstruction quality is worse, and the volume of the reconstructions is sometimes inconsistent.

**Softmax vs. ReLU**   Using a softmax nonlinearity in the encoder with an entropy-based diversity loss is clearly inferior (#1.5 and #1.6), and the reconstruction quality also reflects this.

Table 2: Results for ADAs trained on code sequences produced by model #1.2. NLLs are reported in nats per timestep. The asterisks indicate our preferred architectures which we use for further experiments.

| # | MODEL | HOP SIZE | DECODER RF | RECONSTRUCTION NLL TRAIN | EVAL | CODEBOOK PERPLEXITY |
|---|---|---|---|---|---|---|
| 2.1 | WaveNet | N/A | 6144 | 4.415 | 4.430 | N/A |
| *2.2 | VQ-VAE (PBT) | 8 | 64 | 4.183 | 4.191 | 226.0 |
| 2.3 | AMAE | 8 | 32 | 4.337 | 4.348 | 227.7 |
| *2.4 | AMAE | 8 | 64 | 4.236 | 4.247 | 228.0 |
| 2.5 | AMAE | 8 | 128 | 4.134 | 4.143 | 226.2 |
| 2.6 | AMAE | 8 | 256 | 4.153 | 4.159 | 234.2 |

Based on our evaluation, we selected architecture #1.2 as a basis for further experiments. We use this setup to train a model with hop size 64 (#1.7). The reconstruction quality of this model is surprisingly good, despite the $8\times$ larger compression factor.

## 4.2 Sequence predictability

Because an ADA produces discrete code sequences, we can train another ADA on top of them. These code sequences differ from waveforms in interesting ways: there is no ordinal relation between the different discrete symbols. Code sequences are also less predictable locally. This can be shown by training a simple predictive model with increasing RF lengths $r$, and looking at how the NLL evolves as the length increases. Namely, we use a 3-layer model with a causal convolution of length $r$ in the first layer, followed by a ReLU nonlinearity, and then two more linear layers with a ReLU nonlinearity in between. We train this on waveforms and on code sequences produced by model #1.2. The resulting *predictability profiles* are shown in Figure 2.

As expected, the recent past is very informative when predicting waveforms, and we quickly reach a point of diminishing returns. The NLL values for code sequences are on a different scale, indicating that they are much harder to predict. Also note that there is a clear transition when the RF length passes 64, which corresponds exactly to the RF of the encoder of model #1.2. This is no coincidence: within the encoder RF, the ADA will try to represent and compress signal information as efficiently as possible, which makes the code sequence more unpredictable locally. This unpredictability also makes it harder to train an ADA on top from an optimisation perspective, because it makes the learning signal much noisier.

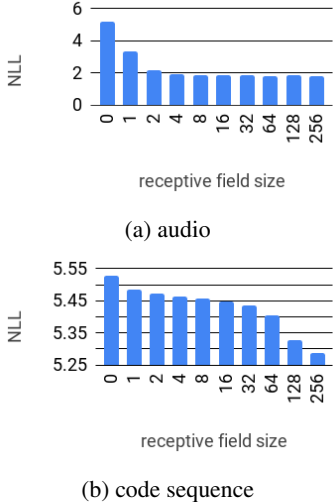

(a) audio

(b) code sequence

Figure 2: Predictability profiles: NLLs obtained by a simple predictive model as a function of its receptive field size. For RF = 0, we estimate the unigram entropy per timestep.

## 4.3 ADA architectures for code sequences

We train several second-level ADAs on the code sequences produced by model #1.2. These models are considerably more challenging to train, and for VQ-VAE it was impossible to find hyperparameters that lead to reliable convergence. Instead, we turn to population based training (PBT) [22] to enable online adaptation of hyperparameters, and to allow for divergence to be detected and mitigated. We run PBT on $\alpha$ and $\beta$ (see Section 3.1). For AMAE models, PBT turns out to be unnecessary, which makes them more suitable for training second-level ADAs, despite performing slightly worse. Results are shown in Table 2.

We find that we have to use considerably smaller decoder RFs to get meaningful results. Larger RFs result in better NLLs as expected, but also seem to cause the reconstructions to become less recognisable. We find that a relatively small RF of 64 timesteps yields the best perceptual results. Note that VQ-VAE models with larger decoder RFs do not converge, even with PBT.

Table 3: Overview of the models we consider for qualitative evaluation, with ratings out of five for signal fidelity and musicalty from an informal human evaluation. We report the mean and standard error across 28 raters' responses.

| # | MODEL | NUM. LEVELS | RF | HUMAN EVALUATION FIDELITY | MUSICALITY |
|---|-------|-------------|-----|---------|------------|
| 3.1 | Large WaveNet | 1 | 384 ms | $3.82 \pm 0.18$ | $2.43 \pm 0.14$ |
| 3.2 | Very large WaveNet | 1 | 768 ms | $3.82 \pm 0.20$ | $2.89 \pm 0.17$ |
| 3.3 | Thin WaveNet with large RF | 1 | 3072 ms | $2.43 \pm 0.17$ | $1.71 \pm 0.18$ |
| 3.4 | hop-8 VQ-VAE + large WaveNet | 2 | 3072 ms | $3.79 \pm 0.16$ | $3.04 \pm 0.16$ |
| 3.5 | hop-64 VQ-VAE + large WaveNet | 2 | 24576 ms | $3.54 \pm 0.18$ | $3.07 \pm 0.17$ |
| 3.6 | VQ-VAE + PBT-VQ-VAE + large WaveNet | 3 | 24576 ms | $3.71 \pm 0.18$ | $4.04 \pm 0.14$ |
| 3.7 | VQ-VAE + AMAE + large WaveNet | 3 | 24576 ms | $3.93 \pm 0.18$ | $3.46 \pm 0.15$ |

## 4.4 Multi-level models

We can train unconditional WaveNets on the code sequences produced by ADAs, and then stack them together to create hierarchical unconditional models of waveforms. We can then sample code sequences unconditionally, and *render* them to audio by passing them through the decoders of one or more ADAs. Finally, we qualitatively compare the resulting samples in terms of signal fidelity and musicality. The models we compare are listed in Table 3.

We have evaluated the models qualitatively by carefully listening to the samples, but we have also conducted an informal blind evaluation study. We have asked individuals to listen to four samples for each model, and to rate the model out of five in terms of fidelity and musicality. The mean and standard error of the ratings across 28 responses are also reported in Table 3.

**Single-level models**    Model #3.1 (which is the same as #1.1), with a receptive field that is typical of a WaveNet model, is not able to produce compelling samples. Making the model twice as deep (#3.2) improves sample quality quite a bit but also makes it prohibitively expensive to train. This model corresponds to the one that was used to generate the piano music samples accompanying the original WaveNet paper [43]. If we try to extend the receptive field and compensate by making the number of units in each layer much smaller (so as to be able to fit the model in RAM, #3.3), we find that it still captures the piano timbre but fails to produce anything coherent.

**Two-level models**    The combination of a hop-size-8 VQ-VAE with a large WaveNet trained on its code sequences (#3.4) yields a remarkable improvement in musicality, with almost no loss in fidelity. With a hop-size-64 VQ-VAE (#3.5) we lose more signal fidelity. While the RF is now extended to almost 25 seconds, we do not find the samples to sound particularly musical.

**Three-level models**    As an alternative to a single VQ-VAE with a large hop size, we also investigate stacking two hop-size-8 ADAs and a large WaveNet (#3.6 and #3.7). The resulting samples are much more interesting musically, but we find that signal fidelity is reduced in this setup. We can attribute this to the difficulty of training second level ADAs.

Most samples from multi-level models are harmonically consistent, with sensible chord progressions and sometimes more advanced structure such as polyphony, cadences and call-and-response motives. Some also show remarkable rhythmic consistency. Many other samples do not, which can probably be attributed at least partially to the composition of the dataset: romantic composers, who often make more use of free-form rhythms and timing variations as a means of expression, are well-represented.

The results of the blind evaluation are largely aligned with our own conclusions in terms of musicality, which is encouraging: models with more levels receive higher ratings. The fidelity ratings on the other hand are fairly uniform across all models, except for #3.3, which also has the poorest musicality rating. However, these numbers should be taken with a grain of salt: note the relatively large standard errors, which are partly due to the small sample size, and partly due to ambiguity in the meanings of 'fidelity' and 'musicality'.

# 5 Discussion

We have addressed the challenge of music generation in the raw audio domain, by using autoregressive models and extending their receptive fields in a computationally efficient manner. We have also introduced the argmax autoencoder (AMAE), an alternative to VQ-VAE which shows improved stability on our challenging task. Using up to three separately trained autoregressive models at different levels of abstraction allows us to capture long-range correlations in audio signals across tens of seconds, corresponding to 100,000s of timesteps, at the cost of some signal fidelity. This indicates that there is a trade-off between accurate modelling of local and large-scale structure.

The most successful approaches to audio waveform generation that have been considered in literature thus far are autoregressive. This aspect seems to be much more important for audio signals than for other domains like images. We believe that this results from a number of fundamental differences between the auditory and visual perception of humans: while our visual system is less sensitive to high frequency noise, our ears tend to perceive spurious or missing high-frequency content as very disturbing. As a result, modelling local signal variations well is much more important for audio, and this is precisely the strength of autoregressive models.

While improving the fidelity of the generated samples (by increasing the sample rate and bit depth) should be relatively straightforward, scaling up the receptive field further will pose some challenges: learning musical structure at the scale of minutes will not just require additional model capacity, but also a lot more training data. Alternative strategies to improve the musicality of the samples further include providing high-level conditioning information (e.g. the composer of a piece), or incorporating prior knowledge about musical form into the model. We look forward to these possibilities, as well as the application of our approach to other kinds of musical signals (e.g. different instruments or multi-instrumental music) and other types of sequential data.

**Acknowledgments**

We would like to thank the following people for their help and input: Lasse Espeholt, Yazhe Li, Jeff Donahue, Igor Babuschkin, Ben Poole, Casper Kaae Sønderby, Ivo Danihelka, Oriol Vinyals, Alex Graves, Nal Kalchbrenner, Grzegorz Świrszcz, George van den Driessche, Chris Jones, Guillaume Desjardins, Georg Ostrovski, Will Dabney, Francesco Visin, Alex Mott, Daniel Zoran, Danilo Rezende, Jeffrey De Fauw, James Besley, Chloe Hillier and the rest of the DeepMind team. We would also like to thank Jesse Engel, Adam Roberts, Curtis Hawthorne, Cinjon Resnick, Sageev Oore, Erich Elsen, Ian Simon, Anna Huang, Chris Donahue, Douglas Eck and the rest of the Magenta team.

## Footnotes

[1]Musical Instrument Digital Interface

[2]$[x]$ is implemented as `tf.stop_gradient(x)` in TensorFlow.

[3]$f(x_i) = \frac{\mathrm{ReLU}(x_i)}{\sum_j \mathrm{ReLU}(x_j) + \epsilon}$

[4]The log-likelihood of a signal (using the natural logarithm) is divided by the number of timesteps for easy comparison across different sequence lengths.

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
