[Supplementary Material]

# A  Details of model architecture and training

The autoregressive discrete autoencoders that we trained feature a local model in the form of a WaveNet, with residual blocks, a multiplicative nonlinearity and skip connections as introduced in the original paper [43]. The local model features 32 blocks, with 4 repetitions of 8 dilation stages and a convolution filter length of 2. This accounts for a receptive field of 1024 timesteps or 64 ms. Within each block, the dilated convolution produces 128 outputs, which are passed through the multiplicative nonlinearity to get 64 outputs (*inner block size*). This is then followed by a length-1 convolution with 384 outputs (*residual block size*).

The encoder and modulator both consist of 16 such residual blocks (2 repetitions of 8 dilation stages) and use non-causal dilated convolutions with a filter length of 3, resulting in a receptive field of 512 timesteps in both directions. They both have an inner block size and residual block size of 256. The encoder produces 8-bit codes (256 symbols) and downsamples the sequence by a factor of 8. This means that the receptive field of the encoder is 32 ms, while that of the modulator is 256 ms. To condition the local model on the code sequence, the modulator processes it and produces time-dependent biases for each dilated convolution layer in the local model.

The 'large' WaveNets that we trained have a receptive field of 6144 timesteps (30 blocks, 3 repeats of 10 dilation stages with filter length 3). They have an inner block size and a residual block size of 512. The 'very large' WaveNet has a receptive field of 12288 timesteps instead, by using 60 blocks instead of 30. This means they take about 4 times longer to train, because they also have to be trained on excerpts that are twice as long. The thin WaveNet with a large receptive field has 39 blocks (3 repeats of 13 dilation stages), which results in a receptive field of 49152 timesteps. The residual and inner block sizes are reduced from 512 to 192 to compensate.

The models are trained using the Adam update rule [24] with a learning rate of $2 \cdot 10^{-4}$ for 500,000 iterations (200,000 for the unconditional WaveNets). All ADAs were trained on 8 GPUs with 16GB RAM each. The unconditional WaveNets were trained on up to 32 GPUs, as they would take too long to train otherwise. For VQ-VAE models, we tune the commitment loss scale factor $\beta$ for each architecture as we find it to be somewhat sensitive to this (the optimal value also depends on the scale of the NLL term). For AMAE models, we find that setting the diversity loss scale factor $\nu$ to $0.1$ yields good results across all architectures we tried. We use Polyak averaging for evaluation [35].

When training VQ-VAE with PBT [22], we use a population size of 20. We randomly initialise $\alpha$ from $[10^{-4}, 10^{-2}]$ and $\beta$ from $[10^{-1}, 10]$ (log-uniformly sampled), and then randomly increase or decrease one or more parameter values by 20% every 5000 iterations. No parameter perturbation takes place in the first 10000 iterations of training. The log-likelihood is used as the fitness function.

# B  Dataset

The dataset consists of just under 413 hours of clean recordings of solo piano music in 16-bit PCM mono format, sampled at 16 kHz. In Table 4, we list the composers whose work is in the dataset. The same composition may feature multiple times in the form of different performances. When using live recordings we were careful to filter out applause, and any material with too much background noise. Note that a small number of recordings featured works from multiple composers, which we have not separated out. A list of URLs corresponding to the data we used is available at https://bit.ly/2IPXoDu. Note that several URLs are no longer available, so we have only included those that are available at the time of publication. We used 99% of the dataset for training, and a hold-out set of 1% for evaluation.

Because certain composers are more popular than others, it is easier to find recordings of their work (e.g. Chopin, Liszt, Beethoven). As a result, they are well-represented in the dataset and the model may learn to reproduce their styles more often than others. We believe a clear bias towards romantic composers is audible in many model samples.

Table 4: List of composers whose work is in the dataset.

| COMPOSER | MINUTES | PCT. | COMPOSER | MINUTES | PCT. |
|---|---|---|---|---|---|
| Chopin | 4517 | 18.23% | Medtner | 112 | 0.45% |
| Liszt | 2052 | 8.28% | Nyman | 111 | 0.45% |
| Beethoven | 1848 | 7.46% | Tiersen | 111 | 0.45% |
| Bach | 1734 | 7.00% | Borodin | 79 | 0.32% |
| Ravel | 1444 | 5.83% | Kuhlau | 78 | 0.31% |
| Debussy | 1341 | 5.41% | Bartok | 77 | 0.31% |
| Mozart | 1022 | 4.12% | Strauss | 75 | 0.30% |
| Schubert | 994 | 4.01% | Clara Schumann | 74 | 0.30% |
| Scriabin | 768 | 3.10% | Haydn / Beethoven / Schumann / Liszt | 72 | 0.29% |
| Robert Schumann | 733 | 2.96% | Lyapunov | 71 | 0.29% |
| Satie | 701 | 2.83% | Mozart / Haydn | 69 | 0.28% |
| Mendelssohn | 523 | 2.11% | Vorisek | 69 | 0.28% |
| Scarlatti | 494 | 1.99% | Stravinsky / Prokofiev / Webern / Boulez | 68 | 0.27% |
| Rachmaninoff | 487 | 1.97% | Mussorgsky | 67 | 0.27% |
| Haydn | 460 | 1.86% | Rodrigo | 66 | 0.27% |
| Einaudi | 324 | 1.31% | Couperin | 65 | 0.26% |
| Glass | 304 | 1.23% | Vierne | 65 | 0.26% |
| Poulenc | 285 | 1.15% | Cimarosa | 61 | 0.25% |
| Mompou | 282 | 1.14% | Granados | 61 | 0.25% |
| Dvorak | 272 | 1.10% | Tournemire | 61 | 0.25% |
| Brahms | 260 | 1.05% | Sibelius | 55 | 0.22% |
| Field / Chopin | 241 | 0.97% | Novak | 54 | 0.22% |
| Faure | 214 | 0.86% | Bridge | 49 | 0.20% |
| Various composers | 206 | 0.83% | Diabelli | 47 | 0.19% |
| Field | 178 | 0.72% | Richter | 46 | 0.19% |
| Prokofiev | 164 | 0.66% | Messiaen | 35 | 0.14% |
| Turina | 159 | 0.64% | Burgmuller | 33 | 0.13% |
| Wagner | 146 | 0.59% | Bortkiewicz | 30 | 0.12% |
| Albeniz | 141 | 0.57% | Reubke | 29 | 0.12% |
| Grieg | 134 | 0.54% | Stravinsky | 28 | 0.11% |
| Tchaikovsky | 134 | 0.54% | Saint-Saens | 23 | 0.09% |
| Part | 120 | 0.48% | Ornstein | 20 | 0.08% |
| Godowsky | 117 | 0.47% | Szymanowski | 19 | 0.08% |
| TOTAL | 24779 | 100.00% | | | |