[Reviews · NeurIPS 2018]

Reviewer 1



Building on the WaveNet-inspired paradigm of neural audio generation, the authors consider how to generate longer duration audio. This is an unsolved problem at present, with many architectures unable to encode all timescales of audio structure in neural representations. The method performs a kind of hierarchical autoencoding, with fine timescales being summarised and then re-generated by a next layer. The authors are thus able to train a system that generates tens of seconds of raw audio waveform. The feasibility of training is a contribution of the work. The authors argue correctly that listening is currently the best way to evaluate generated raw audio in these settings. The audio examples are interesting and persuasive, with good general structure and some interesting quirks. The paper discusses these well. The author response includes results of an additional listening test, which is useful. In summary, I find the paper is a useful practical step forward in neural nets for audio generation, incremental on the theoretical side, but a good demonstration in practice.

Reviewer 2



This paper presents work on autoregressive discrete autoencoders, which are models that are well-suited for modeling long-range temporal structure and generating audio sequences. The authors present a set of tips and tricks that enable learning for such a problem, and in the process they introduce a new type of quantizing VAE. This paper is very well-written. It is easy to follow, there is a sense of purpose in the narrative, and many obvious questions are properly addressed. Kudos to the authors for not simply listing what worked, and actually motivating their approach. The resulting outputs are definitely an improvement as compared to the usual deep-AR models, exhibiting better consistency across time. I was, however, a little disappointed that the definition of "long-term" and "high-level" used in this paper only spanned a few seconds. I would argue that these terms encompass composition form (e.g. ABA structure), and not so much the content from the last few seconds. As with many such papers, I will agree with the authors that coming up with metrics is often pointless since listening is probably the best way to evaluate things. This of course creates a problem in evaluating significance of this work, although informally I found the submitted soundfiles to be pretty good. I would perhaps detract a few points since I don't see a wider applicability to this work. I would be more convinced had I seen a bit more complex music (in terms of timbres), or some glimpse that this work is relevant outside of this specific problem. This is however a solid paper, and I'd be happy to see it accepted.

Reviewer 3



The authors claim that there is no suitable metric to evaluate the quality of the generated audio, which is plausible, so they listened to the audio and evaluated on their own. The only shortcoming here is that no systematic and blind listening test has been conducted yet. The authors themselves might be biased and thus, the capabilities of the proposed approach cannot be considered as fully proven from a scientific perspective. However, a link to the audio is provided so that the readers can convince themselves from the proposed method. Minor comments: -"nats per timestep": should be defined -p. 3, l. 115-116: The reviewer is not sure, whether the term "hop size" is suitable to describe the ratio between the sample rates. -p. 7, l. 278: For AMEA models, PBT turns (a comma helps the reader) -References: [12], [22], [43] are using "et al." even though the author list is not much longer. All authors should be named to be consistent. Quality: The authors address a prevailing research topic and has some novel contributions (see below) to the field of audio and music generation. The topic is going along with the recent advances in deep architectures for audio generation. Clarity: The paper is well written and the proposed methods are explained carefully. A good description of the state-of-the-art in audio generation and their shortcomings are included. The references are complete. Originality: To the knowledge of the reviewer, in this contribution, music is modelled directly on the waveform for the first time, modelling also long-term dependencies related to consistent musical expression and style. The conditioning signal is learnt with an autoencoder. Moreover, argmax autoencoder is proposed as an alternative (faster and more robust) for the quantisation step employed in vector quantisation variational autoencoders, avoiding the "codebook collapse" problem. Significance: A systematic evaluation is missing (see above). However, a link to audio samples is provided in the paper and the proposed approach is well-founded. Confidence: The reviewer is confident with audio processing and machine learning/deep learning and has many peer-reviewed publications in journals and conference proceedings in this field.